# Mapping the single-cell landscape of acral melanoma and analysis of the molecular regulatory network of the tumor microenvironments

Zan He[1†], Zijuan Xin[2,3,4†], Qiong Yang[2,3,4†], Chen Wang[2,3,4†], Meng Li[2,3,4], Wei Rao[1], Zhimin Du[1], Jia Bai[1], Zixuan Guo[5], Xiuyan Ruan[2,3,4], Zhaojun Zhang[2,3,4], Xiangdong Fang[2,3,4]*, Hua Zhao[1]*

[1]Department of Dermatology, The First Medical General Hospital of People's Liberation Army, Beijing, China; [2]Beijing Institute of Genomics & China National Center for Bioinformation, Chinese Academy of Sciences, Beijing, China; [3]University of Chinese Academy of Science, Beijing, China; [4]Beijing Key Laboratory of Genome and Precision Medicine Technologies, Beijing, China; [5]Senior Department of Orthopedics, The Fourth Medical Center of General Hospital of People's Liberation Army, Beijing, China

*For correspondence:
fangxd@big.ac.cn (XF);
hualuck301@163.com (HZ)

[†]These authors contributed equally to this work

**Abstract** Acral melanoma (AM) exhibits a high incidence in Asian patients with melanoma, and it is not well treated with immunotherapy. However, little attention has been paid to the characteristics of the immune microenvironment in AM. Therefore, in this study, we collected clinical samples from Chinese patients with AM and conducted single-cell RNA sequencing to analyze the heterogeneity of its tumor microenvironments (TMEs) and the molecular regulatory network. Our analysis revealed that genes, such as *TWIST1*, *EREG*, *TNFRSF9*, and *CTGF* could drive the deregulation of various TME components. The molecular interaction relationships between TME cells, such as MIF-CD44 and TNFSF9-TNFRSF9, might be an attractive target for developing novel immunotherapeutic agents.

## Editor's evaluation

Historically, the study of acral melanoma has been neglected due to the low proportion it represents out of all melanoma cases among European-descent individuals, which has translated into an important gap of knowledge in the field and hindered the development of effective therapies to control the disease. Therefore, studies that address this unmet need in melanoma research are very important. Here, He and collaborators analyse eight samples from six patients with acral melanoma through single-cell RNA sequencing. They describe the tumour microenvironment in these tumours, including descriptions of interactions among distinct cell types in the tumour microenvironment and potential biomarkers. This study will help inform our knowledge of the immune infiltration on this type of cancer, and is an important step toward better understanding how these cell interactions influence acral melanoma development, progression and therapy response.

## Introduction

Malignant melanoma is a malignant tumor associated with the melanocytes. In recent years, the incidence of malignant melanoma has been increasing, with an annual increase rate of 3-5%. Approximately

**eLife digest** Acral melanoma is a type of cancer that affects the hands and feet. It tends to form on the palms, soles, and under the nails. It is rare in people of European descent, but in Asian populations it makes up more than half of all melanoma cases. Unlike other types of skin cancer, it does not respond well to immunotherapy, but scientists did not understand why.

Historically, cancer research has focused on the genetics of whole tumors. But cancer is complicated. Malignant cells recruit other cells to help them survive and grow, and to protect them from attacks by the immune system. Together, they create their own ecosystem, called the tumor microenvironment. The exact makeup of the tumor microenvironment differs depending on the type of cancer and on the genetics of the individual. Investigating the cells that 'support' the tumor could help to explain how acral melanoma develops and why it does not respond to treatment.

To address these questions, He et al. collected samples from six patients with acral melanoma and examined the genes used by more than 60,000 individual cells. This revealed nine different types of cells in the tumor microenvironment. Most were cancer cells, but there were also immune cells, blood vessel cells, skin cells, and a type of cell that makes connective tissue. He et al. also identified four genes that most likely shape the tumor microenvironment, and two gene pairs that may control some of the interactions between the cells.

Investigating these early findings in more detail could open new treatment avenues for acral melanoma. The number of samples in this study was small, but it provides a starting point for future investigation. With more data, researchers could start to develop treatments that target the unique tumor microenvironment of this type of cancer.

300,000 new cases and 60,000 deaths are recorded every year. It poses a serious threat to the health of people (*Swick and Maize, 2012*). For patients with confirmed or suspected in situ melanoma, surgical resection is the first treatment option, with a good prognosis provided that metastasis had not occurred. However, metastatic melanoma is the most fatal type of cancer (*Wahid et al., 2018*), beceuse patients are susceptible to relapse after surgery, resulting in a poor prognosis and a 5-year survival rate of <10% (*McKean and Amaria, 2018*). Melanoma is generally divided into four types: cutaneous melanoma (CM; occurring primarily in the head, neck, trunk, and limbs), acral melanoma (AM; occurring primarily in the palms, soles of feet, and nail beds where there are no hair follicles), mucosal melanoma, and uveal melanoma. In contrast to the epidemiological data of a proportion of <10% AMs out of the total number of cases of melanoma in the European-descent populations, the proportion of AM cases of the total melanomas in Asians is as high as 70% (*Chi et al., 2011*; *Cormier et al., 2006*), and terminal AMs have been considered as a subtype of melanoma with poor prognosis and poor immune efficacy (*Nakamura et al., 2020*; *Rose et al., 2021*).

The invasion and metastasis of malignant tumors are the pathological basis of tumor recurrence, disease deterioration, and eventual death. The development of a tumor, including invasion and metastasis, is a continuous, progressive, multifactor, and multistep process in coordination with the tumor microenvironments (TMEs). The immune system recognizes tumor antigens and kills tumor cells, but it is not sufficiently strong to eliminate tumors that have formed within the body. Solid tumors are complex tissues comprising not only tumor cells but also stromal cells, inflammatory cells, the vascular system, and the extracellular matrix (ECM), which are collectively defined as the TMEs (*Wan et al., 2013*).

Single-cell sequencing is a study of the genome and transcriptome at the single-cell level (*Tang et al., 2009*). Through genome-wide or RNA amplification, high-throughput sequencing can elucidate the gene structure and gene expression status of individual cells, reflecting the intercell heterogeneity. Compared with the traditional RNA sequencing, the single-cell RNA sequencing (scRNA-seq) is more suitable for analyzing the TME components and heterogeneous populations. Its application in the investigations of the TME has provided an unprecedented solution to its cellular and molecular complexities; thus, deepening our understanding of the heterogeneity, plasticity, and complex crossover interactions among different cell types in the TMEs. With the continuous accumulation of scRNA-seq datasets, it will become an indispensable component of tumor immunology, and it will continue to drive scientific innovations in precision immunotherapy and eventually be adopted in

**Table 1.** Clinical information of samples.

| Patient | Sample | Code name | Gender | Age | Site | Size | Medical history | Diagnose |
|---|---|---|---|---|---|---|---|---|
| Patient 1 | Primary lesion | PL1 | M | 40 | Fourth toe of right foot | 1*2 cm | 20 years, ulcer 1 month | Invasive melanoma |
| | Primary lesion | PL2 | | | Fourth toe of right foot | 5*8 cm | | Malignant melanoma of the extremity |
| Patient 2 | Lymph gland | LG2 | M | 65 | Left foot heel | 2.6 cm in diameter | 5 years, ulcer 6 months | Lymph gland metastasis of melanoma |
| Patient 3 | Adjacent tissue | AT3 | F | 52 | The left groin | 3*2 cm | 8 months, ulcer 2 months | Acral melanoma, T4b (Breslow >4 mm, ulcer) |
| | Primary lesion | PL4 | | | | 4.5*4.5 cm | | Acral melanoma, (Breslow = 2.8 mm) |
| Patient 4 | Adjacent tissue | AT4 | F | 72 | Right foot heel | NA | 2 years | |
| Patient 5 | Primary lesion | PL5 | F | 77 | Left thumb | 3*2 cm | 10 years | Acral melanoma, ulcer |
| Patient 6 | Adjacent tissue | AT6 | F | 69 | Left foot heel | NA | 7 months | Invasive melanoma, (Breslow >1.8 mm) |

routine clinical practice in the foreseeable future. This new technique allows us to conduct better characterization of developmental lineages and differentiation states, which are crucial in understanding the underlying mechanisms that drive the functional diversity of the immune cells in the TMEs.

In this study, we collected lesion biopsy samples from six patients with clinical AM along with three adjacent paracancerous tissues and a metastatic lymph gland (LG) sample and performed 10× Genomics scRNA-seq and analyses. Our study delineates the scRNA landscape of AM and describes the molecular regulatory network of the TME cells. Combined with cytological experiments, we validated some of the results of our analyses, providing an important reference for AM research.

## Results

### scRNA-seq and cell type identification revealed the heterogeneity of AM

We collected eight clinical tissues from six patients with AM, including four samples of primary lesions (PLs), three samples of adjacent tissues (ATs), and one sample of LG metastatic tissue. The clinical characteristics and pathological information of the patients are presented in *Table 1*. After quality control and removal of batch effects, a total of 61,726 single cells were used for downstream analyses (*Figure 1A* and *Figure 1—figure supplement 1A, B*). Gao et al. developed an integrative Bayesian segmentation approach called copy number karyotyping of aneuploid tumors (CopyKAT) to estimate genomic copy number profiles at an average genomic resolution of 5 Mb from read depth in high-throughput scRNA-seq data. According to the copy number variation (CNV) results by CopyKAT (*Gao et al., 2021*), we initially identified aneuploid mutant cells as malignant cells, which were derived specifically from primary and LG metastatic tissues, and diploid mutant cells were identified as cells of the other microenvironmental components (*Figure 1—figure supplement 1C–E*). The microenvironment component cells were clustered into 15 clusters and annotated using SingleR (*Aran et al., 2019*; *Figure 1—figure supplement 1F–H*). All cells were defined as malignant cells (*MITF*[+], 21,624 cells), cancer-associated fibroblasts (CAFs; *COL1A1*[+], 17,434 cells), T cells (*CD3D*[+], 8322 cells), macrophages (*C1QB*[+], 5520 cells), endothelial cells (*PECAM1*[+], 4033 cells), neutrophils (*S100A8*[+], 2519 cells), B cells (*CD79A*[+], 1007 cells), epithelial cells (*EPCAM*[+], 657 cells), and keratinocytes (*KRT14*[+], 607 cells) (*Figure 1B–E*). We counted the number of cells per sample and calculated the ratios of the cell types (*Figure 1F*).

To discover the specific microenvironmental characteristics of AMs, we compared published scRNA-seq data from non-acral skin CM with our sequencing data (GSE115978 and GSE72056) (*Jerby-Arnon et al., 2018*; *Tirosh et al., 2016*). In the PLs, the AMs had a higher proportion of malignant

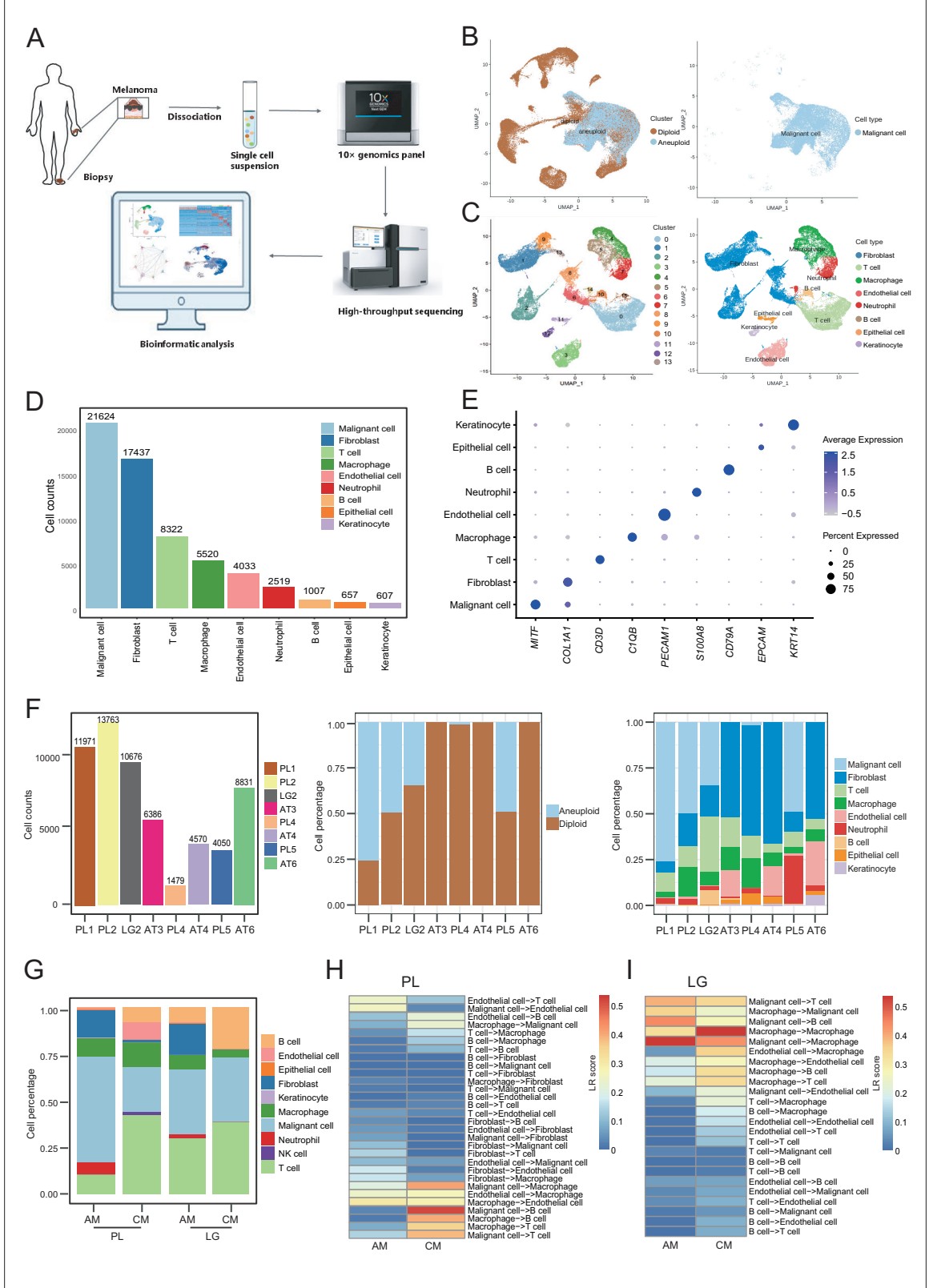

**Figure 1.** Cell composition and heterogeneity in acral melanomas (AMs). (**A**) The technical route of this study, includes sample preparation, sequencing, and bioinformatics analytical process. (**B**) Uniform manifold approximation and projection(UMAP) plot showing the CopyKAT results. The brown dots indicate diploidy and the blue dots indicate aneuploid. The aneuploid cells are considered as malignant, while the diploid cells are considered as stromal cells. (**C**) UMAP plot of nonmalignant cells labeled by cell cluster and cell type. (**D**) Cell count bar plot of each cell type. Malignant cells were the

*Figure 1 continued on next page*

*Figure 1 continued*

most abundant among all cell types and fibroblasts were the most abundant cells among all stromal cells. (**E**) Marker gene expression of each cell type, including dot size and color representing the characteristics of gene expression (pct.exp) and average scaling expression (avg.exp.scale) values. (**F**) The bar plot depicts the cell number and the ratio of cell types per sample after quality control. (**G**) Differences in the proportion of cell types between AMs and cutaneous melanomas (CMs). (**H**) Differences in cell interactions in primary lesions (PLs) of AMs and CMs. (**I**) Differences in cell interactions in LGs of AM and CMs.

The online version of this article includes the following figure supplement(s) for figure 1:

**Figure supplement 1.** Data quality control and cell type annotation.

cells and CAFs, but a lower proportion of lymphocyte infiltrations (*Figure 1G*). This is consistent with the hypothesis that AMs may be caused by friction at the extremities. In the metastatic LGs, the proportion of malignant cells was similar between AMs and CMs, but there was still a larger proportion of CAFs in the metastatic LG of AM, which was almost absent in the metastatic LGs of CMs. The endothelial cells and fibroblasts in PLs of AMs interacted more closely with other types of cells, while the secretion signals of immune and myeloid cells in CMs were more active (*Figure 1H*). Furthermore, the malignant cells in metastatic LG of AM secreted stronger signals to the immune and myeloid cells, while other interactions were weaker (*Figure 1I*).

## Interaction networks among the microenvironments of AM

To determine the cellular components that play key roles in the TME, we delineated the interrelationships among the TME components. According to the gene expression of the receptor-ligand pair, the cell interaction strength within primary tissues, ATs, and LG metastatic tissues was inferred, and the cell interaction network was obtained by CellChat (*Jin et al., 2021*). The results demonstrated that the communications among cells in the primary tissues were closer (*Figure 2A–C*). In the ATs, KIT and WNT cell interaction signaling pathways were specifically identified, in which KIT signals were secreted primarily by endothelial cells and fibroblasts, and WNT signals were primarily secreted by keratinocytes. In the primary tissues, NT, ncWNT, IL1, and GDF cell interaction signaling pathways were specifically identified. NT, ncWNT, and GDF signals were secreted primarily by malignant cells, and IL1 signals were secreted primarily by neutrophils. Chemerin, NRG, and GDF cell interaction signaling pathways were specifically identified in the LG metastatic tissues. Chemerin and NRG signals were mainly secreted by malignant cells, and PSAP signals were primarily secreted by macrophages (*Figure 2D* and *Figure 2—figure supplement 1A–D*). Compared to the PLs, in the LG, the outgoing and incoming interaction capabilities of malignant cells in the TMEs increased, and the interactions among malignant cells and macrophages, B and T cells and fibroblasts increased. These findings suggest that malignant cells, macrophages, B and T cells and fibroblasts play more important roles in the TMEs (*Figure 2E–G* and *Figure 2—figure supplement 1E*). Subsequently, we focused on the subclusters to observe how they play important roles.

## Pseudotime analyses of malignant cells

To distinguish malignant cells according to the extent of malignancy, we designed a pseudotime-based analytical process. The analyses of the malignant cell clusters identified eight cell subclusters at different stages of differentiation. The pseudotime analyses inferred from splicing kinetics showed that cells in subcluster 3 were in an earlier state, whereas cells in subcluster 1 were in a more terminal state (*Figure 3A, B*). Transcription factor (TF) *TWIST1* was highly expressed in terminal site.

Differentially expressed genes (DEGs) of subclusters 1 and 3 were enriched in Gene Ontology (GO) functions. We observed that the highly expressed DEGs in cluster 1 were related to behaviors such as ECM organization, response to transforming growth factor (TGF) β, regulation of epithelial cell proliferation and migration (*Figure 3—figure supplement 1A, B*). We consider that the cells of subcluster 1 are in a higher malignant state. SCENIC can use the random forest to identify the TF coexpression network (*Aibar et al., 2017*). We identified the corresponding TF modules in clusters 3 and 1. Among them, TF *TWIST1* and its corresponding coexpression module in the subcluster 1 had higher area under curve (AUC) scores and expression levels (*Figure 3C* and *Figure 3—figure supplement 1C*). Based on the abovementioned analyses, we suspected that *TWIST1* may play a vital regulatory role in the development of malignant cells by regulating the target genes in its coexpression module (*Figure 3C* and *Figure 3—figure supplement 1D*). We observed that genes in *TWIST1*

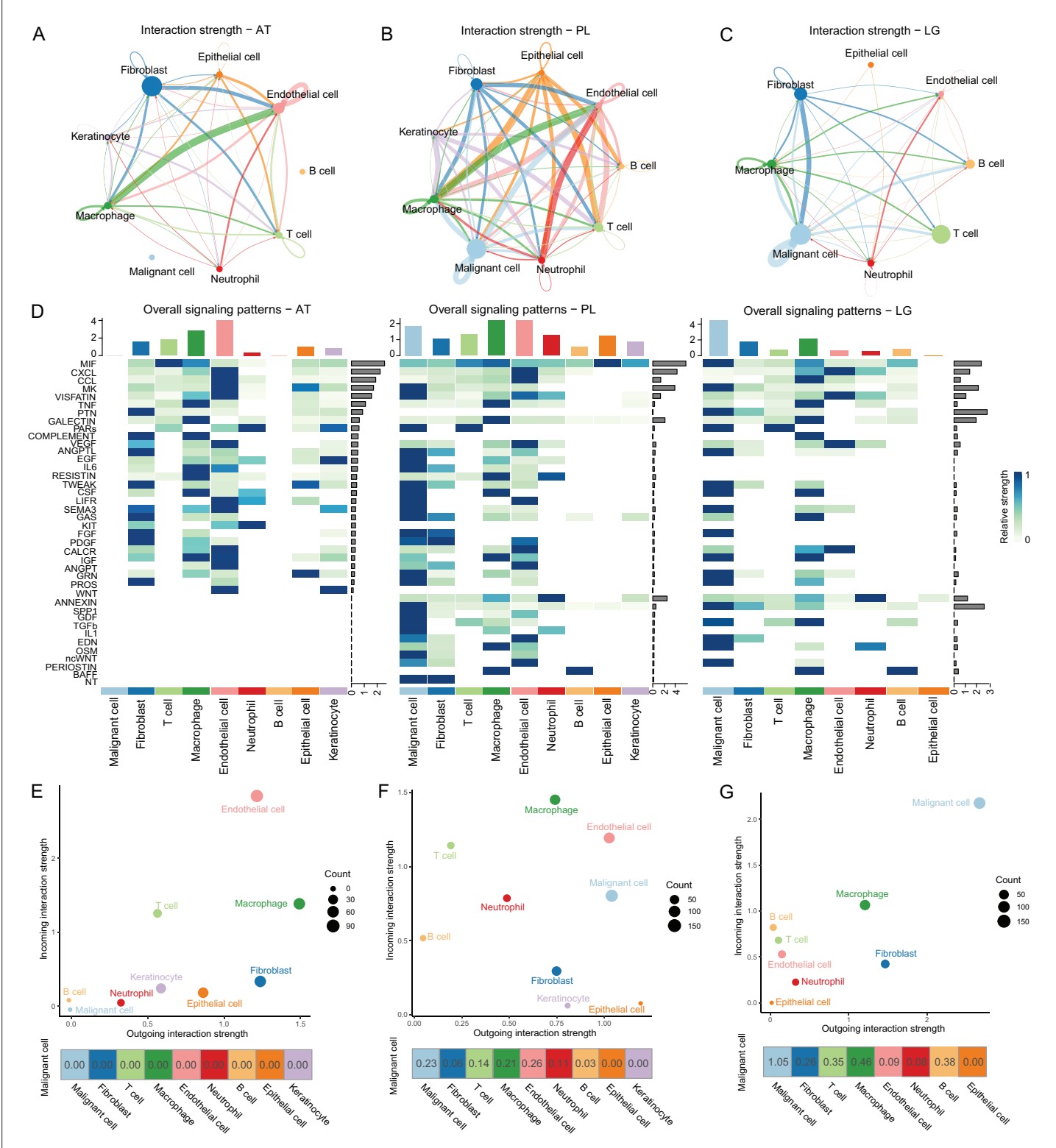

**Figure 2.** Cell-cell communication analysis. (**A**) Adjacent tissue (AT) cell communication network. The interactions between endothelial cells and macrophages were the strongest, while malignant cells did not interact with other types of cells. (**B**) Primary lesion (PL) cell communication network. Different colors represent different cell types. The thickness of the line represents the strength of cell interaction, and the thicker the line, the stronger cell interaction. There are strong and complex interaction signals among different types of cells. (**C**) Lymph gland metastasis sample (LG) cell communication network. There was still the strongest signal interaction between malignant cells and fibroblasts. (**D**) The heatmap shows ATs, primary

*Figure 2 continued on next page*

*Figure 2 continued*

acral melanoma (AM) lesions, and lymph gland metastasis cell interaction pathways identified according to each cell type. The height of the top bar chart represents the interaction strength of each cell type, and the height of the bar on the right represents the strength of the signaling pathway. (**E**) The outgoing and incoming interaction strength of each cell type in ATs. Outgoing means cells secrete signals or have ligands, while incoming means cells receive signals or have receptors. The corresponding outgoing and incoming interaction strength of the cell is obtained through the statistics of expression levels of ligand and receptor coding genes in the cell. The bottom number represents the strength of the malignant cell's interactions with other cell types. (**F**) The outgoing and incoming interaction strength of each cell type in primary AM lesions. (**G**) The outgoing and incoming interaction strength of each cell type in the lymph gland metastasis sample.

The online version of this article includes the following figure supplement(s) for figure 2:

**Figure supplement 1.** Cell-cell interaction network analysis.

target coexpression module contained a variety of collagen-encoding genes such as *MMP2*, indicate that TWIST1 may affect the structures of the ECM by regulating matrix metalloproteinase 2 (MMP2). Further GO functional enrichment disclosed that the genes in the *TWIST1* coexpression module may be involved in biological processes related to tumor degeneration, such as epithelial-mesenchymal transition (EMT) and response to TGF-β (*Figure 3E*). To verify the objectivity of these hypotheses, we transfected the melanoma A375 cell line and performed western blot, wound-healing and Transwell assay (*Figure 3—figure supplement 1E–G*). The assays confirmed that after the transfection of cells with *TWIST1*, EMT pathway proteins were activated, and the invasion and migration abilities of melanoma A375 cells were enhanced.

TWIST1 is one of the EMT-inducer prototypes (*Li et al., 2006*); however, to what extent these different functions of TWIST1 including its effects on EMT, proliferation, and apoptosis are functionally linked or whether these functions are independently regulated by TWIST1 remains unknown (*Gasinska et al., 2018*). We suspected that TWIST1 may influence the interactions between malignant cells and the other cells by regulating ECM. Therefore, we analyzed the interactions between different malignant cell subclusters and other cell types, and showed that the subcluster with high activity of TWIST1 had stronger interaction strength with stroma cells (*Figure 3F*). Cells in subcluster 1 interact with receptors of stromal cells such as fibroblasts, through COL1A1-ITGA2 and other collagen pathways (*Figure 3G* and *Figure 3—figure supplement 1H*), and some of these genes, such as *COL1A1*, *COL6A1*, and *COL6A3*, were regulated by TWIST1 in coexpression module (*Figure 3—figure supplement 1D*). Integrin α2 (ITGA2) triggers cancer cell adhesion to collagen, promotes cell migration, anoikis resistance, mesothelial clearance, and peritoneal metastasis (*Huang et al., 2020*). These results suggest that TF *TWIST1* in malignant cells may promote the secretion of collagen signals and interact with integrin proteins generated by fibroblasts to destroy the integrity of e-cadherin; thus, driving the EMT and metastasis.

## Two different subtypes of macrophages were identified in AM

A reclustering analysis of macrophages distinguished macrophages into two cell types, M1 (*HLA-DQA2*+) and M2 (*CD163*+), among which the tumor-infiltrating macrophages were primarily of the M2 type (*Figure 4A*). Combined with the key motifs identified by SCENIC, we observed that the activities of the three motifs, *IRF4*, *KLF9*, and *SOX18*, were downregulated, and the activation of *STAT1*, *REL*, and *NF-κB1* motifs resulted in this M2 polarization process (*Figure 4—figure supplement 1A*). The cell communication weight demonstrated that there was a strong cell interaction between M2 cells and malignant cells, and there was also a certain cell interaction with M1 cells (*Figure 4B*). Among them, M1 cells primarily secreted tumor necrosis factor (TNF)-α, whereas M2 cells secreted TGF-β (*Wang et al., 2019*). Furthermore, M2 cells affected M1 macrophages through the galectin signaling pathway (*Figure 4C* and *Figure 4—figure supplement 1B*). The heatmap of pseudotime gene dynamic expression inferred from splicing kinetics suggested that *EREG* was highly expressed in M2 cells (*Figure 4D, E* and *Figure 4—figure supplement 1C*). Moreover, patients with melanoma with high expression of *EREG* in The Cancer Genome Atlas (TCGA) cohort had a lower survival rate (*Figure 4F*). We divided the M2-type macrophages into two groups of high and low of *EREG* expression, and the differential analyses revealed that M2 cells with high expression of *EREG* also expressed CD44 (*Figure 4G* and *Figure 4—figure supplement 1D, E*). We describe the molecular interactions among malignant cells, M1 and M2 macrophages. The MIF ligand molecules affected M1 and M2 macrophages through the CD44 receptor. MIF is a macrophage migration inhibitory molecule (*Donn*

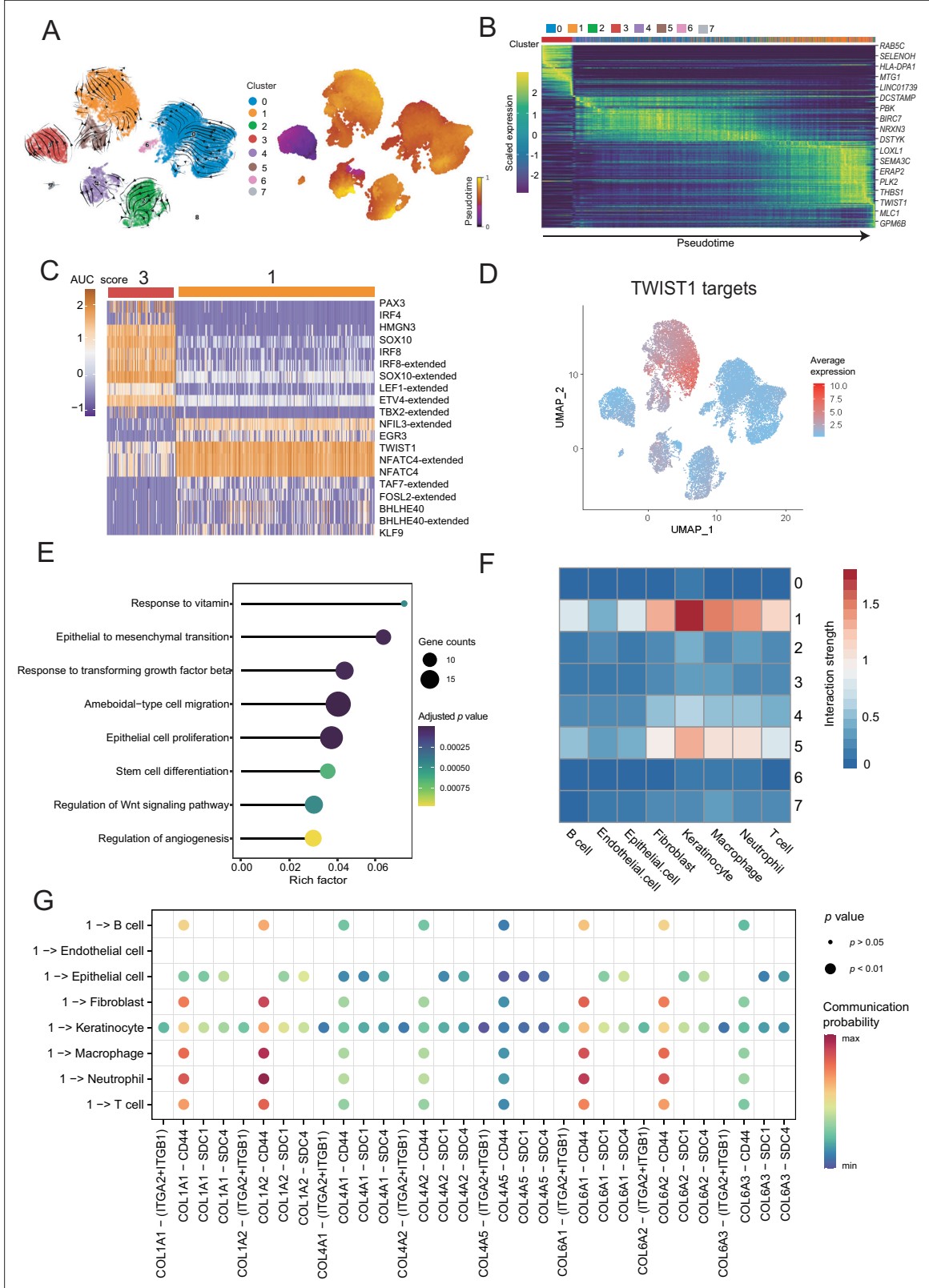

**Figure 3.** Malignant cells acquire the ability to invade presumably via epithelial-mesenchymal transition (EMT) and extracellular matrix (ECM) pathways. (**A**) UMAP plot shows the RNA velocities and latent time of malignant cells. (**B**) Heatmap shows the dynamic gene expression patterns accompanying the evolution of malignant cells. (**C**) Heatmap of the area under curve (AUC) scores of transcription factor (TF) motifs estimated by SCENIC for each cell in subclusters 1 and 3. (**D**) *TWIST1* target expression levels in UMAP plot. (**E**) GO term enrichment results of TWIST1 targets genes in coexpression

*Figure 3 continued on next page*

*Figure 3 continued*

module. (**F**) Heatmap shows the interaction between the different malignant cell subclusters and the other cell types. (**G**) Bubble chart shows ligand-receptor pairs secreted by subcluster 1 to stroma cells.

The online version of this article includes the following source data and figure supplement(s) for figure 3:

**Figure supplement 1.** Malignant cell subcluster analysis.

**Figure supplement 1—source data 1.** Original data records and pictures of Western blot, Wound-healing assay and Transwell assay.

*and Ray, 2004*), which implies that the migration ability of M2 cells with a high expression of *EREG* decreases. Simultaneously, M2 cells with a low expression of *EREG* were enriched in the pathway related to macrophage migration (*Figure 4H* and *Figure 4—figure supplement 1F*).

## Lymphocyte subcluster analysis revealed specific transcriptional characteristics of AM

Lymphocytes are widely believed to play a complex role in the TMEs; hence, we analyzed B and T cells. We divided the B-cell subclusters into naive B (*IGHM*+), memory B (*CD27*+), germinal center B (*BCL6*+), and plasma cells (*CD38*+) (*Figure 5A* and *Figure 5—figure supplement 1A*). The germinal center B cells and plasma cells were detected at the beginning and end of the pseudotime, respectively (*Figure 5—figure supplement 1B*). Furthermore, we discovered the molecular regulatory relationship between various types of B cells and malignant cells (*Figure 5B, C* and *Figure 5—figure supplement 1D*). The germinal center B cells acted on the CD44 receptors of malignant cells through the LGALS9 ligand, whereas the malignant cells acted on the CD74+ CXCR4 and CD74+ CD44 receptors of various types of B cells primarily through the MIF ligand molecules. It has been reported that the MIF/CD74 signaling pathway plays a vital role in maintaining the survival of germinal center B cells and enhancing the antigen presentation of B cells (*Lantner et al., 2007*). The above-described results suggested that B-cell infiltration primarily inhibited cancer cells in AM.

ProjecTILs was used to annotate T cells in the data into nine subclusters, including CD4+ T cells (CD4_NaiveLike, Th1, Tfh, and Treg) and CD8+ T cells (CD8_NaiveLike, CD8_EarlyActiv, CD8_EffectorMemory, CD8_Tpex, and CD8_Tex) (*Figure 5D*). CD4+ T cells were gradually differentiated into Tregs, which corresponded to the characteristic gene set (cluster 3), represented by *FOXP3* (*Figure 5E, F* and *Figure 5—figure supplement 2A, B*). FOXP3 is a key regulator of regulatory T (Treg) cell gene expression, which can activate the expression of *TNFRSF18*, *IL2RA*, and *CTLA4* and inhibit the expression of *IL2* and *IFNG* in association with TF RUNX1 (*Wu et al., 2006*). The pseudotime trajectory depicted the difference in the gene expression of CD8+ T cells from CD8_NaiveLike to exhausted (or pre-exhausted) T cells (CD8_Tpex, CD8_Tex) (*Figure 5G* and *Figure 5—figure supplement 2C*). In the CD8+ T pseudotime gene clustering heatmap, with the direction of pseudotime, the CD8+ T cells gradually transformed from cluster 3 genes with the high expression of *GNLY* and *IGHG1* to cluster 2 genes with the high expression of *GZMK* and *IFNG*. In terminal cluster 1, the exhausted genes represented by *LAG3*, were highly expressed (*Figure 5H*). Treg, CD8_Tpex, and CD8_Tex interact strongly with malignant cells. Mechanistically, malignant cells use MIF and MDK as their major ligands to produce molecular links with various T-cell receptors, which may play an important role in regulating the exhaustion of T cells in the TMEs. Moreover, compared with the well-known immune checkpoints *PDCD1* and *CTLA-4*, the expression of *TNFRSF9* in exhausted T cells is higher. Conversely, the CM T-cell data obtained from a previously published melanoma single-cell dataset that we downloaded, the expression of *PDCD1* and *CTLA-4* was higher than that of TNFRSF9 (*Figure 5—figure supplement 2D–F*). Simultaneously, malignant cells will be more specific to Tregs, CD8_Tpex, and CD8_Tex through the TNFRSF9 receptor expressed by the TNFSF9 ligand, which suggests that TNFRSF9 may be a more suitable immune check target for the treatment of AM (*Figure 5I* and *Figure 5—figure supplement 2G*).

## CTGF+ CAFs were closely associated with prognosis and degeneration

We detected that CAFs accounted for the largest proportion of cells after malignant cells, and there were several interactions with malignant cells, which could play a key role in TMEs. Therefore, we then divided CAFs into three subclusters. Along with the traditional iCAF (*RGS4*+) and mCAF (*PDGFRA*+) cell clusters, we detected a new CAF cluster that highly expressed *CTGF* (*Figure 6A*). Through GO

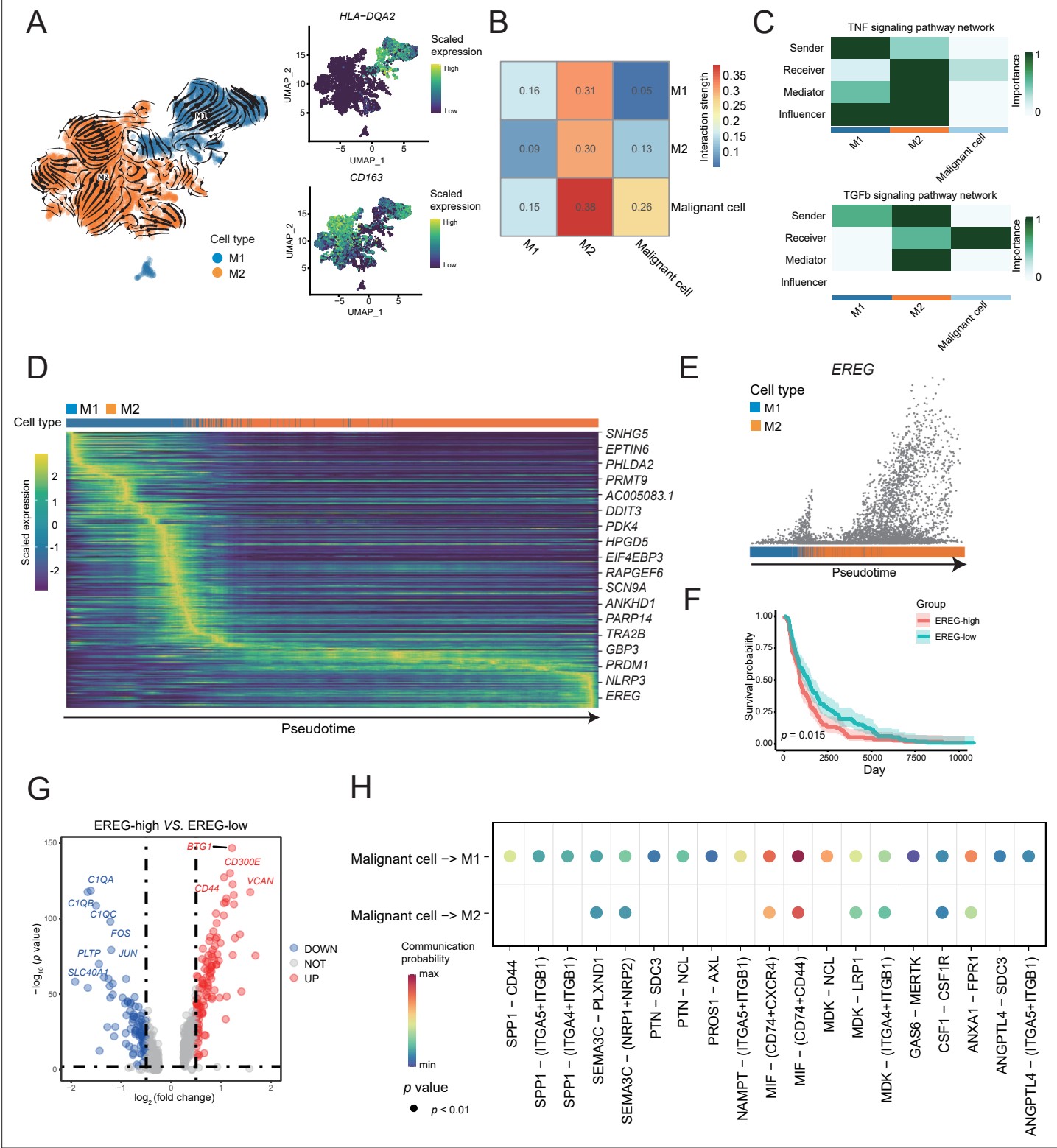

**Figure 4.** M1 and M2 macrophages were found in the macrophage subclusters. (**A**) UMAP plot shows the RNA velocities, HLA-DQA2 and CD163 expression patterns of macrophage subclusters. (**B**) Heatmap of the cell-cell interaction scores was analyzed using CellChat. The M2 cells had the strongest signal interaction with malignant cells. (**C**) The role of each macrophage subcluster cells in the tumor necrosis factor (TNF) and transforming growth factor β (TGF-β) signaling pathways. (**D**) The heatmap shows the dynamic gene expression patterns accompanying the evolution of macrophages. The blue columns represent M1 cells, while the orange columns represent M2 cells. (**E**) Scatter plot shows the *EREG* expression levels

*Figure 4 continued on next page*

*Figure 4 continued*

in macrophages. The direction of arrow is consistent with the direction of pseudotime trajectory. (**F**) Statistics of EREG KM survival curve using the melanoma cohort data in The Cancer Genome Atlas (TCGA). (**G**) The volcano map shows the gene difference analysis between the high- and low-expressing *EREG* cells in the M2 subclusters. (**H**) Bubble chart shows ligand-receptor pairs secreted by the malignant cells to the macrophages.

The online version of this article includes the following figure supplement(s) for figure 4:

**Figure supplement 1.** Macrophage subcluster analysis.

enrichment of the DEGs of the three subclusters, the cell functions of each cluster were observed (*Figure 6B*). The iCAF cluster was associated with the BMP signaling pathway, ECM organization, and extracellular structure organization. The mCAF cluster was associated with the functions of muscle cells and tissues. The CTGF⁺ CAF cluster was associated with functions such as cell growth, response to hypoxia, and protein folding. The SCENIC results revealed that the TWIST1 motif was highly activated in iCAFs, and the TBX2 motif was highly activated in mCAFs. Specifically, CTGF⁺ CAFs activate the NFATC4 and SOX10 motifs (*Figure 6C, D*). In addition, the results of SCENIC inferred that NFATC4 was the upstream TF of *CTGF* and NFATC4 target genes were enriched in GO terms, such as mesenchymal cell proliferation and ECM organization. The melanoma patient cohort in TCGA was divided into two clusters according to the NFATC4 target genes expression patterns, and the patient cluster with a high expression of NFATC4 target genes had a lower survival rate (*Figure 6E, F* and *Figure 6—figure supplement 1A, B*). Compared to the PLs, the expression of CTGF⁺ CAFs in the LG increases, which suggests that the secreted protein CTGF plays a vital regulatory role in the metastasis of malignant cells (*Figure 7A–C*). By adding 50 ng/ml CTGF during the A375 culture process, we observed that CTGF significantly increased the colony formation and cell migration ability of A375 cells (*Figure 7D, E*). Moreover, *CTGF*⁺ CAFs may further impact tumor-associated macrophages (TAMs) by the M2-type macrophages through ligand-receptor molecules, such as SPP1-CD44, PTN-NCL, and MDK-NCL (*Figure 7F*).

## Discussion

In recent years, immunotherapy has emerged as the most popular research field in oncology; thus, addressing the issue of variation in immunotherapy efficacy due to differences in race and tumor types has emerged as a research frontier. Unlike the epidemiological characteristics of AM in the Western population, wherein the proportion of AMs is <10% out of the total number of cases of melanoma, the proportion of AMs in Asian melanoma patients is approximately 70%, accompanied with poor prognosis and immune efficacy. Single-cell sequencing technology can more intuitively reveal the composition of various types of cells in the microenvironments and determine the specific subgroups that cannot be considered by traditional bulk sequencing.

Tumor-infiltrating immune cells, including lymphocytes, TAMs, and myeloid-derived suppressor cells (MDSCs), are important components of the TMEs. On the one hand, these immune cells can kill tumor cells (such as CD8⁺ T cells and NK cells); on the other hand, they can also promote tumor development (*Hanahan and Weinberg, 2011*). Existing studies show that immune cells in the TMEs can play a role in promoting tumor development through immunosuppression (*Joyce and Fearon, 2015*; *Ruffell et al., 2010*), promoting angiogenesis (*DeNardo et al., 2008*), inhibiting apoptosis (*Chen et al., 2011*), secreting growth factors (*Balkwill et al., 2005*), helping tumor cells escape growth suppressing factors (*Lu et al., 2011*), promoting tumor metastasis (*Kessenbrock et al., 2010*), and altering the energy metabolism (*Buck et al., 2017*). Like macrophages, TAMs have two forms of macrophages, M1-type and M2-type. The M1-type macrophages can inhibit and phagocytose tumor cells, whereas the M2-type macrophages play immunosuppressive and tumor-promoting roles. We observed that the M2-type macrophages with a high expression of *EREG* had a weaker migration ability, but their ability to promote tumor development is stronger; thus, creating an immunosuppressive microenvironment.

In addition to immune cellular components, stromal components such as endothelial cells and fibroblasts are important components of the TMEs. By secreting TGF-β (*Noma et al., 2008*) and vascular endothelial growth factor (VEGF) (*Olofsson et al., 1999*), tumor cells can induce and activate CAFs and endothelial cells, change the tumor cell phenotype, reshape the ECM, help generate blood vessels as well as lymphatic vessels, and then accelerate the outward escape of tumor cells (*Folkman,*

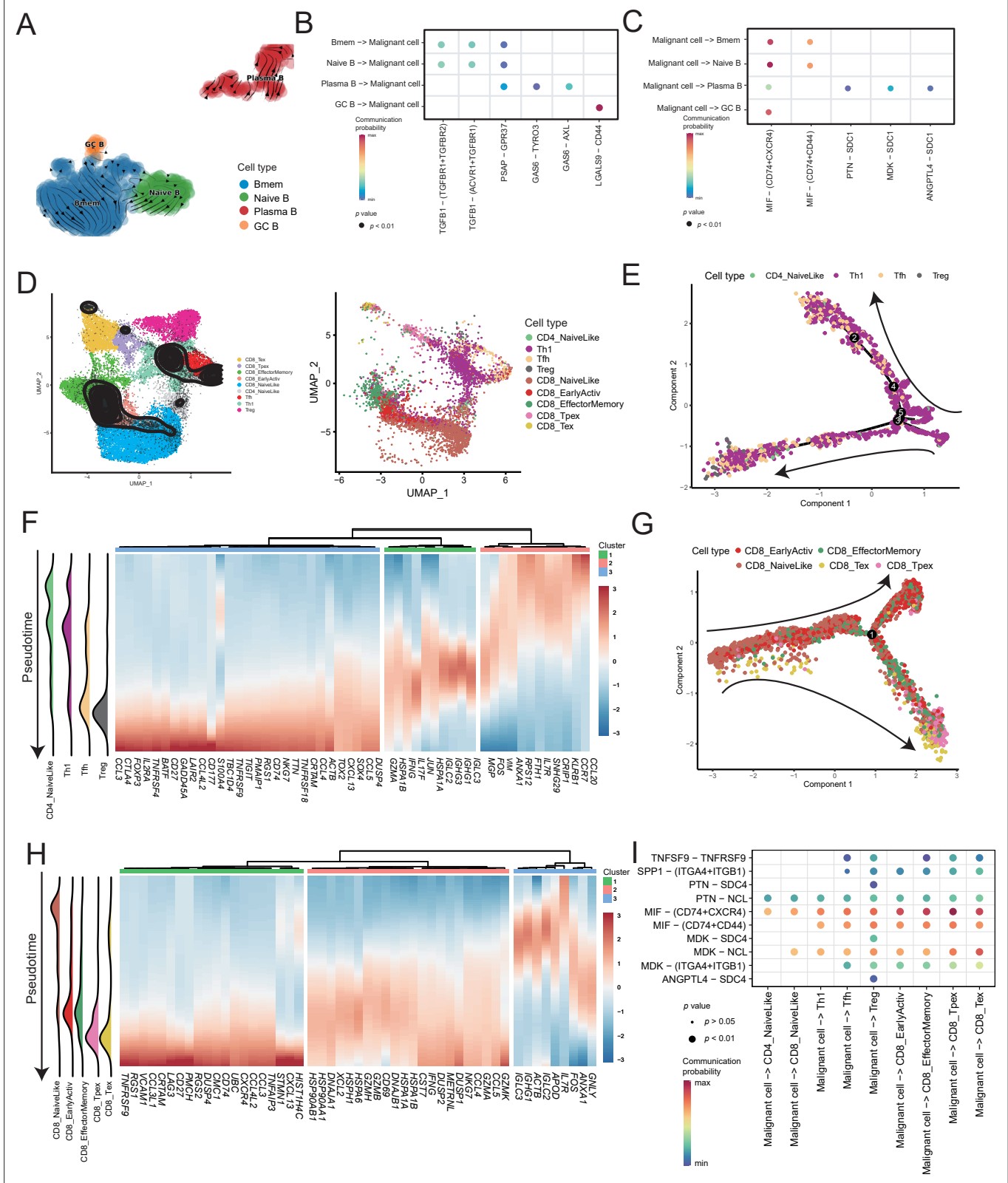

**Figure 5.** Temporal tracing reveals specific transcriptional characteristics of lymphocyte cell subclusters. (**A**) UMAP plot shows the RNA velocities and latency time of the B-cell subclusters. (**B**) Bubble chart showing ligand-receptor pairs secreted by B cells to malignant cells. (**C**) Bubble chart showing ligand-receptor pairs secreted by malignant cells to B cells. (**D**) UMAP map shows the results of T-cell subcluster cell annotation by ProjecTILs. The cells in the colored background refer to the cell types in the reference dataset, and the cells in the black circle represent the cells in our data. (**E**) The

*Figure 5 continued on next page*

*Figure 5 continued*

arrangement of different CD4$^+$ T subcluster cells on the pseudotime trajectory. (**F**) The heatmap shows dynamic gene expression patterns accompanying the differentiation of CD4$^+$ T cells. (**G**) The arrangement of the different CD8$^+$ T subcluster cells on the pseudotime trajectory. (**H**) The heatmap shows dynamic gene expression patterns accompanying the differentiation of CD8$^+$ T cells. (**I**) Bubble chart showing ligand-receptor pairs secreted by malignant cells to T cells.

The online version of this article includes the following figure supplement(s) for figure 5:

**Figure supplement 1.** B-cell subcluster gene expression patterns.

**Figure supplement 2.** T-cell subcluster gene expression patterns.

*1990*). In recent years, researchers have found that the stromal components in the TMEs also affect the antitumor immune effects, indicating that it is important to target the related stromal cells in the process of tumor immunotherapy (*Turley et al., 2015*). Fibroblasts normally maintain the structure of the tissues. However, in the early stage of tumor formation, several chemokines (IL-6, IL-8, etc.) secreted by tumor cells can transform normal fibroblasts around the tumor into CAFs (*Zhang et al., 2021*), gradually forming a microenvironment suitable for malignant proliferation and metastasis of tumor cells.

Given the important role of TMEs in tumor progression, it has become an important therapeutic target. Considering the elimination of immunosuppressive CD8$^+$ T cells in the TME as an example (*Topalian et al., 2016*), antibodies that can block cytotoxic T-lymphocyte antigen-4 (CTLA-4) and programmed cell death-1 (PD-1) have been marketed and have achieved remarkable results in the treatment of melanoma (*Eroglu et al., 2015*), lymphoma (*Robert et al., 2015*), Merkel cell carcinoma (*Engels, 2019*), and other tumors. Targeted drugs that block the activation of the VEGF signaling pathway and inhibit tumor metastasis, such as sorafenib and bevacizumab, have also been widely developed. Nevertheless, although these drugs have improved survival in patients with advanced-stage disease, their efficacy is limited, and the response rate is low for AM, which is more common in the Asian population. Therefore, a better understanding of the TME of AM could accelerate the discovery of new targets or combination therapeutic strategies to help clinicians select the appropriate treatment regimens and predict outcomes. Here, we have generated a single-cell transcriptome landscape, elucidated the components of the microenvironment within AM tissues, and analyzed the cell interactions.

In this study, we identified nine cell types in the AM microenvironment, among which malignant cells accounted for the largest proportion, followed by CAFs. Through cell communication analyses, we found that among ATs, PLs, and LG samples, the interactions between malignant cells and macrophages, T and B cells and CAFs became increasingly stronger. This suggests that metastatic/malignant cells may have a stronger ability to remodel the surrounding stromal cells. We also found that malignant cells with a high expression of *TWIST1* have the highest degree of malignancy and higher ability to invade and metastasize. TWIST1 is one of the EMT-inducer prototypes (*Li et al., 2006*); however, to what extent these different functions of TWIST1, including its effects on EMT, stemness, proliferation, and apoptosis, are functionally linked or whether these functions are independently regulated by TWIST1 remains unknown (*Gasinska et al., 2018*). We suspected that TWIST1 may influence the interactions between malignant cells and other cells by regulating ECM.

Currently, immune therapy primarily targets T cells (PD1/PD-L1 and CTLA-4). However, these T cells with high expressions of *PDCD1* and *CTLA4* are almost absent in a significant number of patients with AM, which may make the patients unresponsive to immune checkpoint suppression therapy. At the transcriptional level, our data found that *TNFRSF9* was highly expressed in CD8$^+$ T cells in the AM microenvironment, and existed at the end of the pseudotime trajectory, while *PDCD1* and *CTLA4* expression levels were relatively low. Additionally, there was a ligand-receptor relationship with tumor cells. Several previous studies have found that TNFRSF9 plays an important role. TNFRSF9 is an important activated immune checkpoint molecule on surface of T cells (*Cheuk et al., 2004*; *Eckstrum and Bany, 2011*), which has a complex bidirectional signal regulation mechanism. The two proteins, TNFRSF9 and TNFSF9, play multiple roles in a variety of cancers, autoimmune, infectious and inflammatory diseases, mediating complex immune responses (*Shuh et al., 2013*). In recent years, many studies have found that TNFRSF9/TNFSF9 is involved in immune regulation of various tumors, including pancreatic cancer and hepatocellular carcinoma (*Glorieux and Huang, 2019*; *Wang et al., 2010*), but their specific role in tumor development has not been clarified. Currently, the prevailing

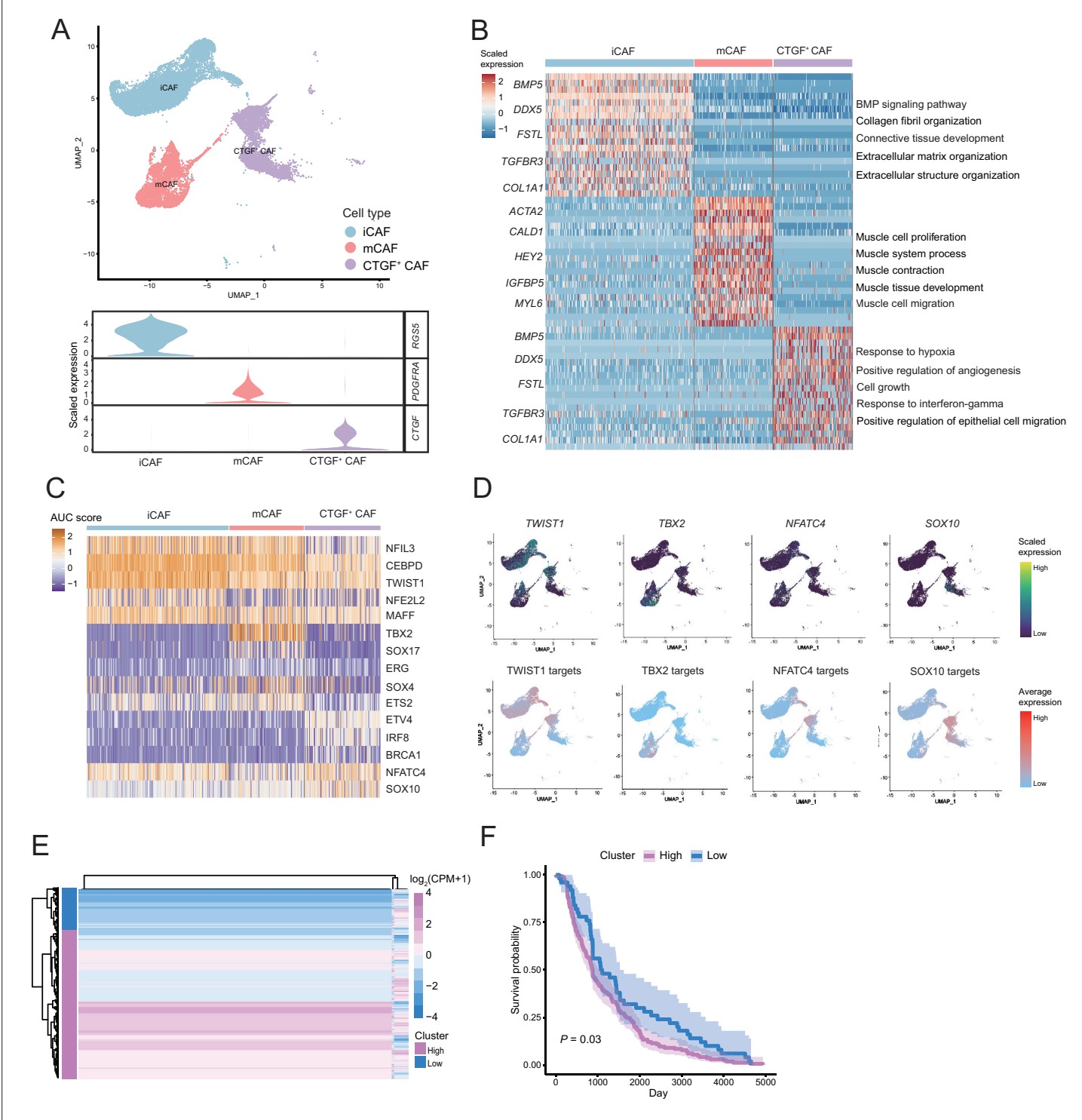

**Figure 6.** The fibroctyeibrocyte subclusters contain three cell types, among which CTGF⁺ cancer-associated fibroblasts (CAFs) are correlated with the degree of malignancy. (**A**) UMAP plot of fibroblast subclusters and Vlnplot of fibroblast subclusters marker genes. (**B**) Heatmap shows the subclusters of differentially expressed genes (DEGs) and enrichment to GO terms. (**C**) Heatmap of the area under curve (AUC) scores of transcription factor (TF) motifs estimated by SCENIC for each cell in fibroblast subclusters. (**D**) UMAP plot of TFs, including *TWIST1, TBX2, NFATC4, SOX10*, and TF target expression levels, in each fibroblast. (**E**) *NFATC4*-targeted expression patterns in melanoma cohort data from The Cancer Genome Atlas (TCGA). (**F**) Statistics of *NFATC4*-targeted KM survival curves using the melanoma cohort data in TCGA.

The online version of this article includes the following figure supplement(s) for figure 6:

**Figure supplement 1.** The targets of NFATC4.

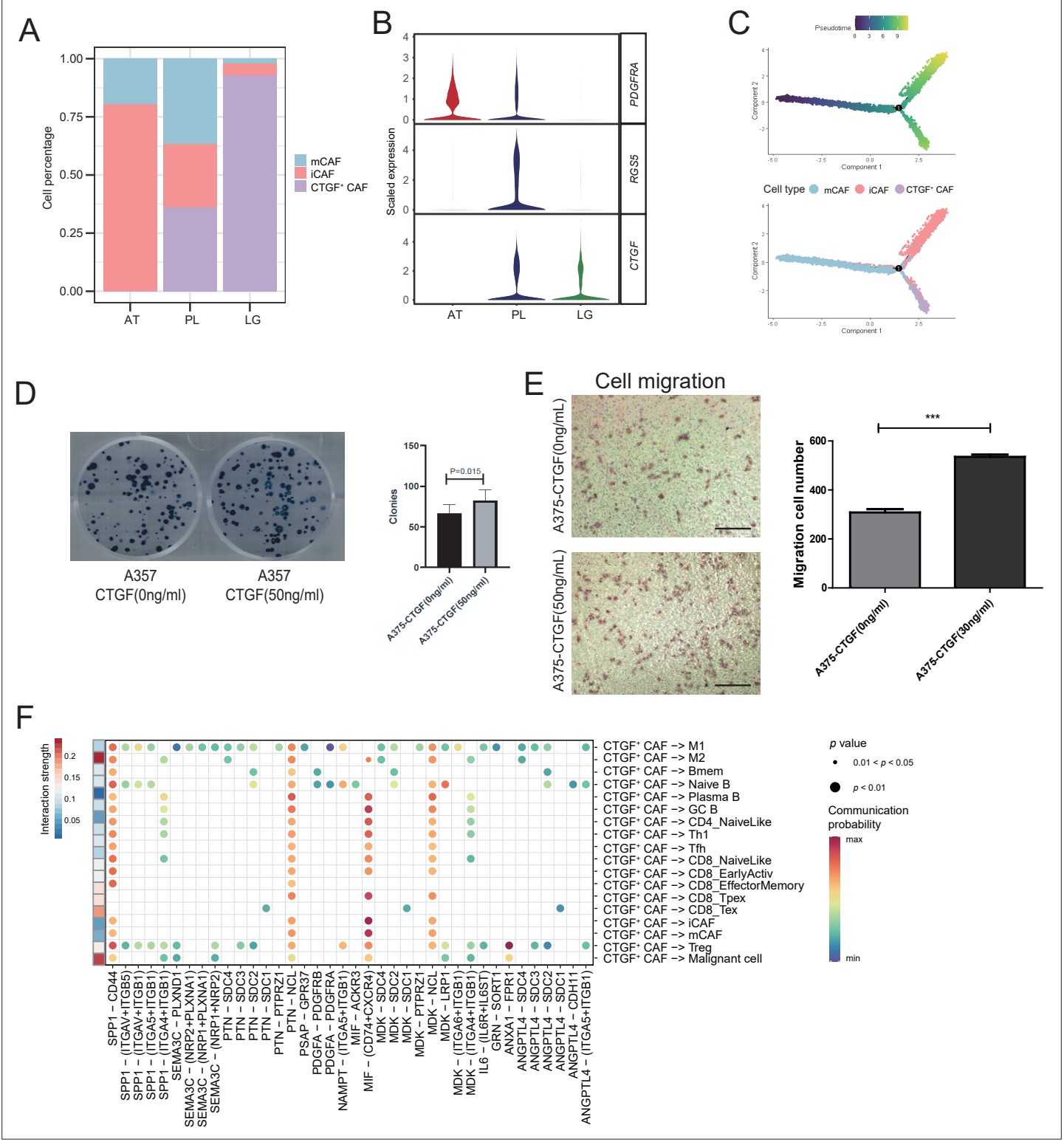

**Figure 7.** *CTGF⁺* cancer-associated fibroblasts (CAFs) promote the proliferation of malignant cells. (**A**) The distribution of fibroblast subclusters in adjacent tissues, primary acral melanoma (AM) lesions, and lymph gland metastasis samples. (**B**) Fibroblast subcluster marker expression levels in adjacent tissues, primary AM lesions, and lymph gland metastasis samples. (**C**) Evolutionary pseudotime of fibroblasts and the arrangement of different subclusters of cells on the pseudotime trajectory. (**D**) Clone formation assay of A375 cells in the presence of 50 ng/ml CTGF. The bar plot depicts the number of clones. Scale bar, 20µm. (**E**) Transwell assay on A375 cells in the presence of 50 ng/ml CTGF. The bar plot shows the number of migrated cells. ***p value < 0.001. (**F**) Interactions of the CTGF⁺ CAFs with the other cell types.

*Figure 7 continued on next page*

*Figure 7 continued*

The online version of this article includes the following source data for figure 7:

**Source data 1.** Original records and pictures of Clone formation assay.

**Source data 2.** Original records and pictures of Transwell assay.

theory is that TNFRSF9 agonists can be used to treat cancers and autoimmune diseases mainly mediated by CD4[+] T cells, but may worsen autoimmune diseases mediated by CD8[+] T cells. Geuijen et al. found that TNFRSF9 agonists in combination with PD-L1 effectively activated and amplified tumor-specific cytotoxic T cells; thus, enhancing tumor control and elimination(*Geuijen et al., 2021*). Therefore, targeting TNFRSF9[+] T cells may be a novel choice for AM therapy but more validations are required to determine whether it is exciting or antagonistic.

Furthermore, CAFs are important in the TMEs. This is because previous studies have shown that CAFs regulate angiogenesis by producing proangiogenic factors, such as FGF-2 and VEGF-A, thereby providing essential information for highly proliferating tumor cells. CAFs can also help tumor cells overcome immune surveillance by recruiting immunosuppressive cells, such as M2 macrophages and MDSCs (*Flavell et al., 2010*; *Yang et al., 2016*). In this AM dataset, we annotated a new subcluster of fibroblasts, CTGF[+] CAFs, which might be a key factor in AM tumor occurrence and metastasis. CTGF[+] CAFs act on malignant cells through CTGF and may occur through SPP1-CD44, PTN-NCL, and MDK-NCL interactions to affect M2-type macrophages. Using AUC of the TF motif score of each cell, as estimated by SCENIC, we found that the NFATC4 motif was specifically activated in CTGF[+]CAFs and TF NFATC4 could activate the expression of genes, such as CTGF. The public dataset shows that patients with a high expression of NFATC4 target genes have lower survival rates. These results suggest that NFATC4 is a potentially important regulator of AM development.

Almost in the same period as our study, Li et al. also carried out a study on scNA-seq for AM, and reported similar, as well as different, results compared to our study (*Li et al., 2022*). For example, we both found out that AMs are less infiltrated by immune cells than non-AMs. However, the two studies identified different microenvironmental components. For instance, our study identified a large proportion of CAFs, which were not obvious in Li's data. Furthermore, we both found novel immunotherapeutic targets that are more suitable for AM but are not samilar. These differences may be caused by the fact that the samples of the two studies were from different ethnic groups and have different sample types. The small sample sizes included in the two studies may also lead to the differences in the results. Therefore, it is still necessary to further expand the study cohort and strengthen the attention to AMs.

However, there are certain limitations associated with our study. First, the number of clinical samples used for analyses was small, which may limit the generalizability of our results. Second, more metastatic samples could have made our grouping more convincing. Finally, the experiments included in our study did not use AM cell lines, and we could not perform experimental verification of all the genes that were mined, because we could not obtain the AM cell lines and experimental materials in time owing to the COVID-19 epidemic.

To summarize, we determined the expression profile of cellular elements in AM and confirmed the characteristics of these tumor-related elements by scRNA-seq. Through in-depth analysis of interactions among the microenvironments components, we proposed the prognostic markers and therapeutic targets with potential for clinical transformation. Our study provides an understanding of cancer immunology and is an important resource for future drug discovery for AM.

## Materials and methods

### Patients and samples

All samples were obtained from the General Hospital of the People's Liberation Army, Beijing, China. All volunteers signed informed consent prior to sample acquisition. Four primary AM tissues, three paracancerous tissues, and a metastatic LG sample were included in this cohort. All experimental procedures were approved by the Ethics Committee of Chinese PLA General Hospital, Beijing, China (Approval No. S2021-626-01).

## Single-cell suspension preparation

The primary AM tissues, adjacent paracancerous tissues, and metastatic LG tissues were processed immediately after being obtained from patients with AM. Single-cell suspensions with high cell viability (>90%) were prepared using an automatic mild tissue processor (Miltenyi gentleMACS Dissociator). Collagenase (1 mg/ml) and elastase (1 mg/ml) were prepared at a ratio of 1:4 and preheated at 37°C. Each sample was cut into small pieces (<1 mm in diameter). The tissues and digestive juices were added into the C tube or M tube matching the instrument, and the corresponding tube cover was used. Then, the tube cover was inverted and installed on the disintegrator. The gentleMACS program was selected from the menu, the steering, and speed were set, and the Start button was pressed. After the operation, the tubes were removed, the tube covers were opened, and the cell suspensions removed. The cells were washed twice with buffer solution and then resuspended to 800-1200 cells/μl. Cells were stained with Trypan blue or fluorescent reagents and counted using the corresponding counting instruments. Cell suspensions with a cell viability of ≥90% and an aggregation rate of ≤5% were used for sequencing.

## Droplet-based single-cell sequencing

The Chromium Single-Cell 3′ Library and Gel Bead KIT V3 (10× Genomics, 1000075) were used to prepare barcoded scRNA-seq libraries according to the manufacturer's protocol. Single-cell suspensions were loaded onto a Chromium Single-Cell Controller Instrument (10× Genomics) to generate single-cell gel beads in emulsions (GEMs), according to the manufacturer's protocol. Approximately 8000 cells were added to each channel to capture 5000 cells per library. The captured cells were lysed, and the released RNA was barcoded through reverse transcription in individual GEMs. Using an S1000TM Touch Thermal Cycler (BioRad) to reverse transcribe, the GEMs were programmed at 53°C for 45 min and 85°C for 5 min and held at 4°C. cDNA was generated and then amplified, and the quality was evaluated using Agilent 4200. Each library was sequenced on an Illumina NovaSeq 6000 sequencer with a sequencing depth of at least 100,000 reads per cell, and 150 bp (PE150) paired-end reads were generated (performed by CapitalBio, Beijing).

## Raw data processing and quality control

Cell Ranger (version 3.3.0) was used to process the raw data, demultiplex cellular barcodes, map reads to the transcriptome, and downsample reads (as required to generate normalized aggregate data across samples). Raw gene expression matrices with unique molecular identifiers (UMIs) generated by the Cell Ranger were imported into Seurat (v.4.0.0) (*Stuart et al., 2019*). Cells with ≥25% of mitochondrial reads and ≤500 unique genes were considered to be of low quality and removed. DoubletFinder was used to eliminate potential doublets. Finally, 61,726 single cells remained, which were applied in downstream analyses (*Figure 1—figure supplement 1A, B*). After removing the potential batch effect, 20 principal components were used for the corresponding analysis, and UMAP was used for nonlinear dimension reduction and visualization.

## Cell type annotation

The CopyKAT (*Gao et al., 2021*) package was used to detect the CNVs in cells and recognize real cancer cells with default parameters. Aneuploid cells were defined as malignant cells, while diploid cells were annotated by SingleR and classified according to the annotation results (*Aran et al., 2019*). CM T-cell single-cell data were downloaded from GEO (GSE120575). Furthermore, T cells were annotated by ProjecTILs (*Andreatta et al., 2021*).

## Trajectory and RNA velocity analyses

RNA velocity and pseudotime analyses were performed using Monocle2 (*Qiu et al., 2017*) and scVelo (*Qiu et al., 2017*). ScVelo is a Python (v3.9.0)-based computational analysis tool. Other data analyses were conducted in the R (v4.0.0) environment.

## Simultaneous gene regulatory network analyses

SCENIC is a new computational method used in the construction of regulatory networks and in the identification of different cell states from scRNA-seq data (*Aibar et al., 2017*). To evaluate the differences among cell clusters based on TFs or their target genes, SCENIC was performed on all single

cells, and the preferentially expressed regulons were calculated using the Limma package (*Ritchie et al., 2015*). Only the regulons that were significantly up- or downregulated in at least one cluster, with an adjacent p value <0.05, were included in further analysis.

## Cell-cell communication analyses

Cell contact patterns were constructed using CellChat (v.0.0.2) (*Jin et al., 2021*). CellChat uses gene expression data as input and combines the interactions of ligand receptors and their cofactors to simulate cell-to-cell communication. It can identify communication patterns and predict the function of understudied pathways and the key signaling events among spatially colocalized cell populations.

## AM and CM comparisons

The differences in the cell population levels between AM from this study were compared with previously published scRNA-seq data from non-acral skin melanoma. Additional analyses were performed on two publicly available non-acral CM scRNA-seq datasets (GSE115978 and GSE72056) (*Jerby-Arnon et al., 2018*; *Tirosh et al., 2016*). Non-acral CM in situ samples and LG samples from cutaneous metastasis were selected for comparison with our PL and LG samples.

## Functional enrichment analyses

The DEGs of cell subgroups were recognized using the FindMarker function provided by Seurat. Log$_2$|FC| > 0.5 and adjacent p value <0.01 were used as the cutoff criteria. ClusterProfiler was used for GO/KEGG analyses (Gene Ontology/Kyoto Encyclopaedia of Genes and Genomes) (*Ashburner et al., 2000*; *Kanehisa et al., 2010*; *Yu et al., 2012*). GO was used to describe gene functions from three aspects: biological process, cellular component, and molecular function. The KEGG was explored for pathways at a significance level of p < 0.05.

## Survival analyses

The melanoma patient cohort data used for survival analyses were obtained from TCGA, which are deposited on the GDC website (https://portal.gdc.cancer.gov/repository). The R package(3.2) survival was used to identify the time, status and groups of cohorts, and to plot Kaplan-Meier curves.

## Cell line and cell culture

A375 cell line (ATCC, Manassas, VA, USA; catalog number: CRL-1619) has been verified by STR profiling and tested negative for mycoplasma contamination. Cells were cultured in Dulbecco's modified Eagle's medium (Gibco, Grand Island, NY, USA) containing 10% fetal bovine serum (AusGeneX, Molendinar, Qld, Australia) and 1% penicillin-streptomycin (P/S; Invitrogen, Carlsbad, CA, USA) in a humidified atmosphere in a 5% $CO_2$ incubator at 37°C.

## Transwell migration and invasion assays

For Transwell migration assays, $2 \times 10^4$ cells were resuspended in a serum-free medium and seeded in the upper chamber of the Transwell (8 mm, BD Biosciences, Franklin Lakes, NJ, USA). Medium containing 20% serum was added into the lower chamber. After 18 hr of incubation at 37°C and 5% $CO_2$, the cells remaining in the upper chamber were removed using a wet cotton swab. Transwell invasion assays were performed as described for the migration assays, except that the Transwell upper chamber was coated with Matrigel matrix (dilution 1:7; 356234, BD Biosciences), and $2 \times 10^5$ cells were placed in the upper chamber. The cells that had migrated and adhered to the lower chamber were fixed with 4% paraformaldehyde for 10 min, stained with hematoxylin (ZLI-9610, ZSGB-Bio) and eosin (ZLI-9613, ZSGB-Bio) for 20 min, and then imaged. The number of cells was counted in ten separate high-power fields with vertical cross distribution.

## Western blotting

The adherent cells were dissociated and centrifuged, and the supernatant was collected to prepare protein samples. Protein content was quantified using the bicinchoninic acid (BCA) method (Thermo Scientific 23227). A 5× sodium dodecyl sulfate (SDS) loading buffer was added to the protein sample to dilute it to 1×, and then the protein was placed in a 95°C water bath and heated for 5 min. SDS-polyacrylamide gel electrophoresis (PAGE) gels were prepared, 30-50 μg protein samples were

loaded in each well, and electrophoresis was performed at a constant voltage of 80 V for 10-20 min until all protein samples entered the separation gel completely, and the voltage was adjusted to 120 V. A 0.45-μm polyvinylidene fluoride membrane was selected with a constant current of 300 mA for 2 hr. Subsequently, the primary and secondary antibodies were incubated, and the membranes were exposed in a dark chamber. Grayscale statistics were performed using the One Way ANOVA method. E-cadherin (Cell Signaling Technology, #3195), vimentin (Abcam, AB8069), TWIST1 (Abcam, AB50887), and GAPDH (Abcam, AB75834) antibodies were used.

## Wound-healing assay

At the bottom of a 6-well plate, lines were marked at intervals of 0.5-1 cm, ensuring that at least five lines passed through each well. Cells were dispensed in the 6-well plate at a density of $5 \times 10^5$ cells per well to ensure that the bottom of each plate was covered with a single layer of cells on the next day. The cells were evenly passed along the marking line, and a serum-free medium was added to continue the culture. Samples were collected at 0, 12, 24, and 48 hr, and the site of each photo was located according to the horizontal line and scratched to ensure that cells in the same position were observed.

## Clone formation assay

Cells in the logarithmic growth phase were collected to prepare cell suspensions and counted, the suspension gradient was diluted multiple times, and cells were dispensed at a density of 200 cells per well in a 6-well plate. Three replicates were performed for each cell sample to reduce experimental error. The 6-well plate was placed in a cell incubator for approximately 2 weeks, with the medium being changed every 3 days. The culture was terminated when a population of cells of suitable size appeared at the bottom of the 6-well plate. The supernatant was discarded, and 4% paraformaldehyde solution was added to fix the cells. After removing the paraformaldehyde, an appropriate amount of crystal violet dye solution was added to stain the cells for 5-15 min. Then, the dye solution was slowly washed with running water and air-dried. The 6-well plate was placed on the scanner, and the number of cell clones was counted for statistical analyses.

## Acknowledgements

This research was supported by the National Natural Science Foundation of China (Grant No. 81672698). Funding: This research was supported by the National Natural Science Foundation of China (Grant No. 81672698).

---

## Additional information

### Funding

| Funder | Grant reference number | Author |
| --- | --- | --- |
| National Natural Science Foundation of China | 81672698 | Hua Zhao |

The funders had no role in study design, data collection, and interpretation, or the decision to submit the work for publication.

### Author contributions

Zan He, Conceptualization, Data curation, Validation, Methodology, Writing - original draft, Project administration; Zijuan Xin, Conceptualization, Data curation, Software, Visualization, Methodology; Qiong Yang, Chen Wang, Validation, Methodology; Meng Li, Methodology; Wei Rao, Zhimin Du, Investigation; Jia Bai, Zixuan Guo, Resources; Xiuyan Ruan, Supervision; Zhaojun Zhang, Writing - review and editing; Xiangdong Fang, Conceptualization, Resources, Supervision, Writing - review and editing; Hua Zhao, Conceptualization, Resources, Writing - review and editing

### Author ORCIDs

Xiangdong Fang http://orcid.org/0000-0002-6628-8620
Hua Zhao http://orcid.org/0000-0001-7139-1844

---

## Ethics

All samples were obtained from the General Hospital of the People's Liberation Army, Beijing, China. All volunteers signed informed consent prior to sample acquisition. Four primary AM tissues, three paracancerous tissues, and a metastatic lymph gland sample were included in this cohort. This study was approved by the Ethics Committee of Chinese PLA General Hospital and complied with all relevant ethical regulations (Approval No. S2021-626).

## Decision letter and Author response

Decision letter https://doi.org/10.7554/eLife.78616.sa1
Author response https://doi.org/10.7554/eLife.78616.sa2

# Additional files

## Supplementary files

- MDAR checklist

## Data availability

Sequencing data have been deposited in GSA under accession code HRA001804.

The following dataset was generated:

| Author(s) | Year | Dataset title | Dataset URL | Database and Identifier |
|---|---|---|---|---|
| Zhao Hua | 2022 | Mapping the single-cell landscape of acral melanoma and analysis of the molecular regulatory network of the tumor microenvironment | https://ngdc.cncb.ac.cn/gsa-human/browse/HRA001804 | GSA for Human, HRA001804 |

The following previously published datasets were used:

| Author(s) | Year | Dataset title | Dataset URL | Database and Identifier |
|---|---|---|---|---|
| Jerby-Arnon L, Shah P, Cuoco MS, Rodman C, Su MJ, Melms JC, Leeson R, Kanodia A, Mei S, Lin JR, Wang S, Rabasha B, Liu D, Zhang G, Margolais C, Ashenberg O, Ott PA, Buchbinder EI, Haq R, Hodi FS, Boland GM, Sullivan RJ, Frederick DT, Miao B, Moll T, Flaherty KT, Herlyn M, Jenkins RW, Thummalapalli R, Kowalczyk MS, Cañadas I, Schilling B, Cartwright AN, Luoma AM, Malu S, Hwu P, Bernatchez C, Forget MA, Barbie DA, Shalek AK, Tirosh I, Sorger PK, Wucherpfennig K, Van Allen EM, Schadendorf D, Johnson BE, Rotem A, Rozenblatt-Rosen O, Garraway LA, Yoon CH, Izar B, Regev A | 2018 | Single-cell RNA-seq of melanoma ecosystems reveals sources of T cells exclusion linked to immunotherapy clinical outcomes | https://www.ncbi.nlm.nih.gov/geo/query/acc.cgi?acc=GSE115978 | NCBI Gene Expression Omnibus, GSE115978 |
| Tirosh I, Izar B | 2016 | Single cell RNA-seq analysis of melanoma | https://www.ncbi.nlm.nih.gov/geo/query/acc.cgi?acc=GSE72056 | NCBI Gene Expression Omnibus, GSE72056 |

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
