## [Editor Report]

Historically, the study of acral melanoma has been neglected due to the low proportion it represents out of all melanoma cases among European-descent individuals, which has translated into an important gap of knowledge in the field and hindered the development of effective therapies to control the disease. Therefore, studies that address this unmet need in melanoma research are very important. Here, He and collaborators analyse eight samples from six patients with acral melanoma through single-cell RNA sequencing. They describe the tumour microenvironment in these tumours, including descriptions of interactions among distinct cell types in the tumour microenvironment and potential biomarkers. This study will help inform our knowledge of the immune infiltration on this type of cancer, and is an important step toward better understanding how these cell interactions influence acral melanoma development, progression and therapy response.

---

## [Decision Letter]

**Decision letter after peer review:**

Thank you for submitting your article "Mapping the single-cell landscape of acral melanoma and analysis of the molecular regulatory network of the tumor microenvironment" for consideration by *eLife*. Your article has been reviewed by 2 peer reviewers, including C Daniela Robles-Espinoza as the Reviewing Editor and Reviewer #1, and the evaluation has been overseen by Päivi Ojala as the Senior Editor.

Essential revisions:

In this study, He and collaborators analyse eight samples from six patients with acral melanoma through single-cell RNA sequencing. They describe the tumour microenvironment in these tumours, including descriptions of interactions among distinct cell types and potential biomarkers. The reviewers agreed that this study is important as it focuses on an understudied type of melanoma and is quite detailed. However, they have also identified a number of points to address before publication.

1. In several parts of the text and figures, the authors assume that their samples represent the "progression of tumours" and this seems to be one of the most relevant comparisons that the study makes. However, these samples do not recapitulate tumour progression. Adjacent tissue is not considered a stage of tumour progression. Also, the number of samples is low (especially metastasis n=1) to conclude that changes observed are not sample-dependent. Could the authors please comment on these issues?

2. This study focuses on tumour-microenvironment interactions. When reading such a study, those interested in these interactions and in melanoma biology and treatment will undoubtedly ask if what is seen in acral melanoma is similar to what is seen in non-acral skin melanoma. The only comparison that is made in this sense is regarding TNFRSF9 expression in the current cohort, and in cutaneous melanoma single-cell data previously published. If these comparisons, even if only in the discussion, are not done, it does not help one understand why it would be relevant to study cutaneous melanomas located in acral skin. Could the authors please add a comparison with what has been observed in cutaneous melanoma, are there any differences or similarities between these diseases?

3. The two reviewers commented on this point: Finding that expression of TNFRSF9 is higher than PDCD1 and CTLA-1 in exhausted T cells of acral melanomas is interesting, especially because it is shown that in non-acral cutaneous melanoma, the opposite occurs. But this sole information is not sufficient to suggest that TNFRSF9 would be a "suitable immune check target for AM treatment". Perhaps the authors assume too much from a bioinformatic analysis? Are there any other data that can support this hypothesis? Or is this claim solely based on a biomarker expressed by infiltrating T-cells?

4. Related to "Pseudotime analyses of malignant cells", the main conclusion of this part is that there is a subpopulation of tumour cells (cluster 1) that is in a more terminal/highly malignant state. The authors identify TWIST as a transcription factor that may be related to malignancy and show that it can induce EMT markers and an aggressive phenotype in A375 cells. TWIST is a known EMT regulator with a role in aggressiveness, and therefore this information is not novel. Have the authors considered repeating their experiments in an acral cell line? Considering the line of thought that the authors follow in the other figures, it would be more relevant to look at how TWIST expression influences the communication of melanoma cells with other stromal cells. For instance, is the interaction profile of cells expressing TWIST different from cells expressing transcription factors involved in the Cluster 1 phenotype? Are these populations communicating differently with the stromal cells?

5. Line 85: "we initially identified aneuploid mutant cells as malignant cells, which were derived primarily from primary and lymph gland metastatic tissues, and diploid mutant cells were identified as cells of other microenvironmental components" – what is the basis to assume this? Were most of the diploid cells found in adjacent non-malignant tissue? Were the aneuploid cells expressing oncogenic markers? Can the authors make such an assumption without backing it up with additional evidence?

6. Line 28. "Moreover, a single-cell reference for AM has not been established." Reviewers comment that other work should be acknowledged here, and the authors should seek to place their results in the context of other previous similar work. For example, see similar work by Keiran Smalley's group (PMID 35247927).

7. Finally, both reviewers commented that the figures need much more description to be understandable. Specifically:

a. Figure S1D needs more description than just "Heatmap showing the copycat result". What do the colors and columns mean? Also, please label axes.

b. In Figure S1F, what does a higher score mean?

c. Figure S1G is confusing to a reviewer – It seems that the clusters constituted the immune cell labels (as it appears to be the case in Figure S1F), however, it seems a cluster can be constituted by several different immune cell types? Can the authors specify please what the clusters indicate then?

d. Is the left panel of figure S1G necessary? Is it not redundant with the right panel?

e. Should Figure S3 be called differently? Not all the figure is about TWIST1.

f. Figures 2A-C – do the colors mean anything? What does a thicker line mean? What is an ongoing and incoming interaction strength?

g. Figure 3A – clusters are not labeled so it is not easy to follow the narrative.

h. Figure 3B – What are the colors at the top? The columns?

i. Figure 4A, 4D – labeling colors are important – which color marks M1 and which M2?

j. Figure 5D – What is the black blurb? Can Figure 5F be above Figure 5H? Figures 5F and 5H have the same legend but they don't look the same. Can colours be labeled on top?

---

## [Author Response]

Essential revisions:1. In several parts of the text and figures, the authors assume that their samples represent the "progression of tumours” and this seems to be one of the most relevant comparisons that the study makes. However, these samples do not recapitulate tumour progression. Adjacent tissue is not considered a stage of tumour progression. Also, the number of samples is low (especially metastasis n=1) to conclude that changes observed are not sample-dependent. Could the authors please comment on these issues?

Thank you for good questions. It is not accurate to represent the adjacent tissue samples as ‘tumour progression’, and our study does not want to focus on the tumour developmental process. We have revised the related description in the text.

Tumor adjacent tissues (ATs) have always been the focus of research on TMEs. Some studies believe that there are a lot of mutations and clone amplification in normal tissues adjacent to cancer, which may be in a pre-cancerous state (PMID: 33004515), and many single-cell studies of tumours have also sampled and paired para-cancer tissues (e.g., PMID: 29988129; PMID: 35303421). Metastatic lymph gland (LG) tissue can demonstrate the microenvironmental status of migrated tumour cells in the new environment.

The sample amount does limit the generality of the results, and we pointed out in the Discussion section. Most acral melanoma (AM) patients opt for surgical resection at an early stage to avoid the possibility of metastasis, so we rarely encountered patients with lymph gland (LG) metastases. We only collected one metastatic sample, because it is highly rare in the clinic. The sample that we collected has a high quality, such as a high cell activity of single cell suspension after dissociation (95.30%) and rich densities of tumor cells and other stroma cells. Therefore, we added its sequencing data into the overall analysis, hoping to contribute to the comprehensiveness of resources and research.

2. This study focuses on tumour-microenvironment interactions. When reading such a study, those interested in these interactions and in melanoma biology and treatment will undoubtedly ask if what is seen in acral melanoma is similar to what is seen in non-acral skin melanoma. The only comparison that is made in this sense is regarding TNFRSF9 expression in the current cohort, and in cutaneous melanoma single-cell data previously published. If these comparisons, even if only in the discussion, are not done, it does not help one understand why it would be relevant to study cutaneous melanomas located in acral skin. Could the authors please add a comparison with what has been observed in cutaneous melanoma, are there any differences or similarities between these diseases?

Thanks for your good questions. We have already suppled the comparation of AMs with non-acral skin cutaneous melanomas (CMs), using the published data. The results are supplemented in the first part of the results and Figure 1 G-I. The current results showed that the AMs had a higher proportion of malignant cells and CAFs, but a lower proportion of lymphocyte infiltrations in the primary lesions (PLs). In the metastatic lymph glands (LGs), the proportion of malignant cells was similar between AMs and CMs, but there was still a large proportion of CAFs in the metastatic LGs of AMs, which was almost absent in the metastatic LGs of CMs. The endothelial cells and fibroblasts in PLs of AMs interacted more closely with other types of cells, while the secretion signals of immune cells and myeloid cells in CMs were more active. In LGs, malignant cells in metastatic LGs of AMs secreted stronger signals to immune cells and myeloid cells, and other interactions were weaker.

3. The two reviewers commented on this point: Finding that expression of TNFRSF9 is higher than PDCD1 and CTLA-1 in exhausted T cells of acral melanomas is interesting, especially because it is shown that in non-acral cutaneous melanoma, the opposite occurs. But this sole information is not sufficient to suggest that TNFRSF9 would be a "suitable immune check target for AM treatment". Perhaps the authors assume too much from a bioinformatic analysis? Are there any other data that can support this hypothesis? Or is this claim solely based on a biomarker expressed by infiltrating T-cells?

Thank you for the key question. We agree with you that it is not sufficient to suggest that TNFRSF9 would be a new target. We provided a possibility that TNFRSF9 may serve as a novel therapy. To get more support, it requires multi-omics analyses and assays. In fact, RNA sequencing can only put forward insight from the perspective of expression levels. We perused other similar reports at the scRNA level; for example, a similar article (PMID: 35247927) also considered VISTA and ADORA2 as potential novel targets for immunotherapy according to the increased expression of VISTA and ADORA2. At the transcriptional level, our data found that TNFRSF9 was highly expressed in CD8^+^T cells in the AM microenvironments, and existed at the end of the pseudotime trajectory, whereas PDCD1 and CTLA-4 expression levels were relatively low. In addition, there was a ligand-receptor relationship with tumor cells.

In addition to the differences in expression that transcriptomes can provide, several previous studies have found that TNFRSF9 plays an important role in cancer, autoimmune diseases and inflammation. TNFRSF9 is an important activated immune checkpoint molecule on surface of T cells (PMID: 21560035, PMID: 14671675), which has a complex bidirectional signal regulation mechanism. The two proteins, TNFRSF9 and TNFSF9, play multiple roles in a variety of cancers, autoimmune, infectious and inflammatory diseases, mediating complex immune responses (PMID: 23531747). In recent years, many studies have found that TNFRSF9/TNFSF9 is involved in immune regulation of various tumours, including pancreatic cancer and hepatocellular carcinoma (PMID: 31288851, PMID: 21059892), but their specific role in tumour development has not been clarified. Currently, the prevailing theory is that TNFRSF9 agonists can be used to treat cancers and autoimmune diseases mainly mediated by CD4^+^T cells, but may worsen autoimmune diseases mediated by CD8^+^T cells. Geuijen et al., found that TNFRSF9 agonists in combination with PD-L1 effectively activated and amplified tumour-specific cytotoxic T cells, enhancing tumour control and elimination (PMID: 34290245). Our current results suggested that targeting TNFRSF9 may benefit patients with AM, but more validation is required to determine whether it is activated or anergic. The above content has been supplemented in the sixth paragraph of the Discussion section of our manuscript.

4. Related to "Pseudotime analyses of malignant cells", the main conclusion of this part is that there is a subpopulation of tumour cells (cluster 1) that is in a more terminal/highly malignant state. The authors identify TWIST as a transcription factor that may be related to malignancy and show that it can induce EMT markers and an aggressive phenotype in A375 cells. TWIST is a known EMT regulator with a role in aggressiveness, and therefore this information is not novel. Have the authors considered repeating their experiments in an acral cell line? Considering the line of thought that the authors follow in the other figures, it would be more relevant to look at how TWIST expression influences the communication of melanoma cells with other stromal cells. For instance, is the interaction profile of cells expressing TWIST different from cells expressing transcription factors involved in the Cluster 1 phenotype? Are these populations communicating differently with the stromal cells?

Thank you for your very good question. The pseudotime analyses showed that TWIST1 was highly expressed in terminal site, and SCENIC found that TWIST1 and its corresponding co-expression module in the subcluster 1 had higher AUC score (Figure 3B and Figure3—figure supplement 1C). Therefore, we suspected that TWIST1 might play a vital regulatory role in the development of malignant cells by regulating the target genes in its co-expression module. We have analysed the communication between subcluster1 (*TWIST1*^+^ malignant cells) and other stromal cells, and the results are supplemented in the last paragraph of the third part of our Results section. Figure 3 and Figure3—figure supplement 1 have also been revised. We analysed the interactions between different malignant cell subclusters and other cell types, and showed that the subclusters with high activity of TWIST1 had stronger interaction strength with stroma cells (Figure 3F). Cells in subcluster 1 interacted with receptors of stromal cells, such as fibroblasts through COL1A1-ITGA2 and other collagen pathways (Figure 3G and Figure3—figure supplement 1H), and some of the genes, such as COL1A1, COL6A1, COL6A3, were regulated by TWIST1 in a co-expression module.

We would like to try to replicate the assays in AM cell lines as you rightly suggested, but to our regrets and unfortunately, it is difficult for us to obtain AM cell lines. No commercial AM cell lines can be purchased from ATCC or ECACC. AM cell lines are more difficult to establish and there are few reports on methods for establishing primary acral melanoma cell cultures (PMID: 22578220, PMID: 17488338). Some Japanese and Chinese researchers have isolated the primary generation of AM cells (e.g., PMID: 17488338, PMID: 22578220, PMID: 34097822), but due to the customs policy and the COVID-19 epidemic, we could not receive them within a short period. Moreover, these studies also stated their limitations; namely, that the stability during serial passaging was not evaluated. Therefore, it may be very time-consuming to obtain operable AM cell lines for functional assays. Thus, I am afraid that we would not be able to improve the experiments in a short time; that is really a pity. However, our research group would like to have the opportunity to separate and culture primary cells in subsequent studies, and improve relevant experiments according to your valuable suggestions. Many thanks, again, for your comments.

5. Line 85: "we initially identified aneuploid mutant cells as malignant cells, which were derived primarily from primary and lymph gland metastatic tissues, and diploid mutant cells were identified as cells of other microenvironmental components" – what is the basis to assume this? Were most of the diploid cells found in adjacent non-malignant tissue? Were the aneuploid cells expressing oncogenic markers? Can the authors make such an assumption without backing it up with additional evidence?

Thanks for your question. An effective approach to distinguish tumour cells from normal cells involves the identification of aneuploid copy number profiles, which are common (88%) in most human tumours (PMID: 29622463) and are not found in stromal cell types that have diploid genomes. Gao et al., developed an integrative Bayesian segmentation approach, called copy number karyotyping of aneuploid tumors (CopyKAT), to estimate genomic copy number profiles at an average genomic resolution of 5 Mb from read depth in high-throughput scRNA-seq data (PMID: 33462507). We used this tool to define malignant cells; hence, the term ‘aneuploid’. Adjacent non-malignant tissues barely contain aneuploid cells (Figure1—figure supplement 1E). We also annotated the cells with marker genes; the aneuploid cells were confirmed to express oncogenic markers (Figure 1E and Figure1—figure supplement 1D).

6. Line 28. "Moreover, a single-cell reference for AM has not been established." Reviewers comment that other work should be acknowledged here, and the authors should seek to place their results in the context of other previous similar work. For example, see similar work by Keiran Smalley's group (PMID 35247927).

Thank you for your suggestion. We have cited the reference you mentioned. Both studies were carried out at about the same period, and Li et al.'s study had not yet been published when we wrote our manuscript. We perused the study by Keiran Smalley's Group and found some similarities and differences between the two studies. We have already revised the description in Line 28, and added something new with regard to the research of Keiran Smalley's Group in the Discussion section. For example, we both have found that AMs are less infiltrated by immune cells than non-acral melanomas.

However, the two studies identified different microenvironmental components. We have both found new immunotherapeutic targets that are more suitable for AM, but they are not the same. These differences may be caused by the fact that the samples for the two studies were from different ethnic groups and had different sample types. The small sample sizes included in the two studies may also lead to the difference in results. Therefore, it is still necessary to further expand the study cohort and strengthen the attention to AM.

7. Finally, both reviewers commented that the figures need much more description to be understandable. Specifically:a. Figure S1D needs more description than just "Heatmap showing the copycat result". What do the colors and columns mean? Also, please label axes.

We adjusted Figure S1D to S1C(Figure1—figure supplement 1D to Figure1—figure supplement 1C). Warm colours represent the increase in copy number, while cold colors represent the loss of copy number. Columns represent genomic positions. In aneuploid cells, copy number amplification and deletion are more frequent, which are more consistent with the characteristics of the malignant cells. We have labelled the colour and columns in Figure S1C(Figure1—figure supplement 1C).

b. In Figure S1F, what does a higher score mean?

In Figure S1F (Figure1—figure supplement 1F), higher scores mean higher Spearman correlations of the cells to be annotated with the expression levels of known cells in the reference set.

c. Figure S1G is confusing to a reviewer – It seems that the clusters constituted the immune cell labels (as it appears to be the case in Figure S1F), however, it seems a cluster can be constituted by several different immune cell types? Can the authors specify please what the clusters indicate then?

Figure S1G (Figure1—figure supplement 1G) shows the correspondence between cellular annotation and cellular clusters. Since the cell population information obtained by Seurat process is a method independent of prior knowledge, the same cell cluster generally belongs to the same cell type. At the same time, the number of genes identified by the single-cell sequencing data of 10× Genomics is insufficient, and it is inevitable that there will be fluctuations in the expression of marker genes in a few cells, resulting in false-positive cell annotation results. Therefore, we will take the results of cell classification as the main, and the results of cell annotation as the auxiliary. The proportion of cell types in SingleR annotation results in each cell population will be counted, and the principle of minority obeying majority will be used to determine cell types.

d. Is the left panel of figure S1G necessary? Is it not redundant with the right panel?

According to the answer to question c, both left panel and right panel showed a destabilisation of the annotation tool SingleR. We think the left panel may be not redundant.

e. Should Figure S3 be called differently? Not all the figure is about TWIST1.

Revised. Figure S3(Figure3—figure supplement 1) really should be called separately and we have modified it according to your suggestion.

f. Figures 2A-C – do the colors mean anything? What does a thicker line mean? What is an ongoing and incoming interaction strength?

Revised. Different colours represent different cell types. The thickness of the line represents the strength of cellular interactions, and the thicker the line, the stronger the strength of the interactions. Outgoing means cells secrete signals or have ligands, while incoming means cells receive signals or have receptors. The corresponding outgoing and incoming interaction strength of the cells are obtained through the statistics of expression levels of ligand- and receptor-coding genes in the cells. We have added the above points to the legend of Figure 2.

g. Figure 3A – clusters are not labeled so it is not easy to follow the narrative.

Revised.

h. Figure 3B – What are the colors at the top? The columns?

Revised.

i. Figure 4A, 4D – labeling colors are important – which color marks M1 and which M2?

Revised.

j. Figure 5D – What is the black blurb? Can Figure 5F be above Figure 5H? Figures 5F and 5H have the same legend but they don't look the same. Can colours be labeled on top?

ProjecTILs annotates cell types by comparing the cells to be annotated with the reference data set. In Figure 5D, the cells in the colored background refer to the cell types in the reference dataset, and the cells in the black circle represent the cells in our data. We have modified the position and legends of Figure 5F and H.